# Urinary tract colonization is enhanced by a plasmid that regulates uropathogenic *Acinetobacter baumannii* chromosomal genes

Gisela Di Venanzio [1], Ana L. Flores-Mireles [2,10], Juan J. Calix [3], M. Florencia Haurat[1], Nichollas E. Scott[4], Lauren D. Palmer [5], Robert F. Potter[6], Michael E. Hibbing[2], Laura Friedman[7], Bin Wang[6], Gautam Dantas [1,6,8,9], Eric P. Skaar[5], Scott J. Hultgren [2] & Mario F. Feldman[1]

Multidrug resistant (MDR) *Acinetobacter baumannii* poses a growing threat to global health. Research on *Acinetobacter* pathogenesis has primarily focused on pneumonia and bloodstream infections, even though one in five *A. baumannii* strains are isolated from urinary sites. In this study, we highlight the role of *A. baumannii* as a uropathogen. We develop the first *A. baumannii* catheter-associated urinary tract infection (CAUTI) murine model using UPAB1, a recent MDR urinary isolate. UPAB1 carries the plasmid pAB5, a member of the family of large conjugative plasmids that represses the type VI secretion system (T6SS) in multiple *Acinetobacter* strains. pAB5 confers niche specificity, as its carriage improves UPAB1 survival in a CAUTI model and decreases virulence in a pneumonia model. Comparative proteomic and transcriptomic analyses show that pAB5 regulates the expression of multiple chromosomally-encoded virulence factors besides T6SS. Our results demonstrate that plasmids can impact bacterial infections by controlling the expression of chromosomal genes.

[1] Department of Molecular Microbiology, Washington University School of Medicine, St Louis, MO 63110, USA. [2] Department of Molecular Microbiology, Center for Women's Infectious Disease Research, Washington University School of Medicine, St Louis, MO 63110, USA. [3] Division of Infectious Diseases, Washington University School of Medicine, St. Louis, MO 63110, USA. [4] Department of Microbiology and Immunology, Institute for Infection and Immunity, University of Melbourne at the Peter Doherty, Parkville, Victoria 3010, Australia. [5] Department of Pathology, Microbiology, and Immunology and Vanderbilt Institute for Infection, Immunology and Inflammation, Vanderbilt University Medical Center, Nashville, TN 37232, USA. [6] The Edison Family Center for Genome Sciences and System Biology, Washington University School of Medicine in St. Louis, St. Louis, MO 63110, USA. [7] Universidad de Buenos Aires, Facultad de Farmacia y Bioquímica, Departamento de Microbiología, Inmunología, Biotecnología y Genética, Cátedra de Microbiología, Buenos Aires C1113AAD, Argentina. [8] Department of Pathology and Immunology, Washington University School of Medicine, St. Louis, MO 63110, USA. [9] Department of Biomedical Engineering, Washington University in St. Louis, St. Louis, MO 63105, USA. [10]Present address: Department of Biological Sciences, University of Notre Dame, Notre Dame, IN 46556, USA. Correspondence and requests for materials should be addressed to M.F.F. (email: mariofeldman@wustl.edu)

Human bacterial commensals and obligate pathogens have evolved features that facilitate their survival in a specific habitat. In many cases, these microbial factors are also determinants of virulence and dictate the spectrum of disease associated with the pathogen[1]. For example, some structural features of *Streptococcus pneumoniae* capsule appear to be essential for colonization of the human respiratory tract, but negatively influence survival in blood[2]. In contrast, incidental pathogens with environmental, non-human reservoirs are largely considered to be "niche nonspecific opportunistic pathogens". These pathogens generally cause a wide spectrum of disease dependent on permissive hosts, such as patients that are immunocompromised or critically ill, suffer from breaks in normal immune barriers, or whose microbiomes are perturbed by antimicrobial therapy[3,4]. The Gram-negative bacterium *Acinetobacter baumannii* is generally considered an opportunistic pathogen with no specificity for a particular niche. As a pathogen, it is primarily associated with nosocomial infections, mainly hospital acquired pneumonia, bacteremia, soft tissue infections, and urinary tract infections (UTI)[5], although cases of community acquired infections have been described[6]. Furthermore, *A. baumannii* is recognized as a serious health threat worldwide due to the emerging prevalence of clinical isolates that are multidrug resistant (MDR). Indeed, because *A. baumannii* MDR rates are at least fourfold higher than those for *Pseudomonas aeruginosa*, the second most MDR Gram-negative pathogen[7], the World Health Organization identified *A. baumannii* as a top priority for the research and development of new antimicrobial therapies[8]. However, an incomplete understanding of *A. baumannii* ecology and pathophysiology limits the development of alternative therapeutic strategies.

The two *A. baumannii* strains most commonly used in pathogenesis research, ATCC19606 and ATCC17978[9,10], are non-MDR, lab-domesticated strains that were isolated over 50 years ago. These strains exhibit reduced virulence compared to more recent clinical isolates[11,12] and lack virulence factors identified in modern *A. baumannii* strains, such as the recently described protease CpaA[13]. In order to employ more relevant strains, recent research efforts have adopted contemporary model strains, such as the hypervirulent isolates Ab5075 and LAC-4[14,15]. Under the assumption that pathogenic isolates are equally competent in establishing infection in different anatomical niches in a permissive host, *Acinetobacter* strains are often investigated using infection models that do not match their clinical history. For example, strain Ab5075, isolated in 2008 from a bone infection, has been employed to investigate respiratory infections[14]. *A. baumannii* virulence is principally investigated in vivo using murine pneumonia[15] and sepsis models[16], with only a few reports using soft tissue infection models[17]. Notably, despite early reports highlighting *A. baumannii* as the principal cause of catheter-associated UTI (CAUTI) in some clinical settings[18,19], there is no established model to investigate *A. baumannii* infection in the unique environment of the urinary tract. Thus, current infection models may not be adequate to investigate the full spectrum of *Acinetobacter* disease.

Here, we report that up to one-fifth of *A. baumannii* isolates are obtained from urinary sources, according to a local retrospective study and a systematic review of literature from the last 25 years. To investigate this significant manifestation of *A. baumannii* disease, we develop a murine model of *A. baumannii* CAUTI using a recent MDR UTI isolate, UPAB1. We demonstrate that UPAB1 is able to establish early implant and bladder colonization, dependent on chaperone-usher pathway (CUP) pili. We discovered that UPAB1 harbors a large conjugative plasmid, pAB5, and showed that pAB5 increases UPAB1 virulence in the CAUTI model but is detrimental in a murine pneumonia model.

We linked this behavior to the remarkable ability of pAB5 to impact the expression of multiple chromosomally-encoded virulence factors, such as pili, exopolysaccharides, and protein secretion systems.

## Results

**The urinary tract is a major source of *A. baumannii* isolates.** We performed a retrospective analysis of all *Acinetobacter* isolates identified in the BJC Healthcare System (BJC) from January 2007 through August 2017. Of 2309 identified "*Acinetobacter calcoaceticus-baumannii* complex" (Acbc) cases, 22.2% ($n = 505$) were from urinary sources, compared to 33.9% ($n = 771$), 31.9% ($n = 726$), 10.4% ($n = 237$), and 1.5% ($n = 34$) from respiratory, soft tissue/musculoskeletal (SST/MSK), endovascular, and "other" sources, respectively (Fig. 1). To expand our investigation globally, we performed a systematic review of clinical studies published since 1995, which document the anatomical site of isolation of *A. baumannii* or Acbc specimens. Data were compiled on over 19,000 cases identified as early as 1990, reported in our local study and 12 international clinical studies (Fig. 1, Supplementary Table 1). As in our local findings, respiratory and SST/MSK specimens were the most common isolates worldwide, composing 39.5% and 22.7% of total isolates, respectively. The percentages comprised of urinary isolates varied from study to study (6.1–29.3%), but overall, 17.1% of total isolates (3410 of 19957) were obtained from urinary sources. Thus, despite this clinical relevance, the role of *A. baumannii* as a uropathogen has been largely neglected.

**A CAUTI murine model for *Acinetobacter*.** Approximately one-in-five *A. baumannii* isolates are derived from urinary sources. Though only ~2% of total UTIs are attributed to *Acinetobacter* in global surveillance studies, various single-center studies report that *Acinetobacter* is a leading cause of CAUTI[18], especially in intensive care units[19]. However, there is no model to study *A. baumannii* uropathogenesis. *A. baumannii* infections are commonly associated with the presence of indwelling medical devices, such as endotracheal tubes or central venous lines, and up to 60% of *A. baumannii* urine isolates are from patients with indwelling or intermittent catheters[20]. To simulate this clinical *Acinetobacter* scenario, we adapted a murine CAUTI model frequently used to investigate uropathogenic *Escherichia coli* (UPEC), *Enterococcus faecalis*, methicillin-resistant *Staphylococcus aureus* (MRSA) and Group B *Streptococcus* urinary infections[21,22]. We evaluated the ability of two *A. baumannii* urinary isolates, ATCC19606 ("19606") and UPAB1, to colonize the bladders and kidneys of mice. 19606 is a urinary isolate obtained in 1967 and routinely used to study *A. baumannii* virulence in pneumonia and septicemia murine models[12,23]. In contrast, UPAB1 is an MDR isolate obtained from an ambulatory female patient with an uncomplicated UTI in 2016. Bacteria were inoculated transurethrally after a small piece of silicone tubing (implant) was placed in each mouse urethra. Colonization of the implants, bladders and kidneys was analyzed. Our results show that at 24 h post infection (hpi), 19606 was nearly cleared from infected mice, while bacterial burden increased 5-logs on implants and bladders recovered from UPAB1-infected mice (Fig. 2a). Fluorescence microscopy confirmed the presence of UPAB1 (in red or *Ab*) on the luminal urothelial surface and on the outer surface of silicone implants (Fig. 2b, left panel). Catheterization elicits an inflammatory response in the human bladder, resulting in the release of fibrinogen (Fg) into the bladder lumen, ultimately coating the catheter[24]. Known uropathogens, *E. faecalis* and *S. aureus*, express the Fg binding adhesins, EbpA and ClfB, respectively, that mediate binding and biofilm formation on the

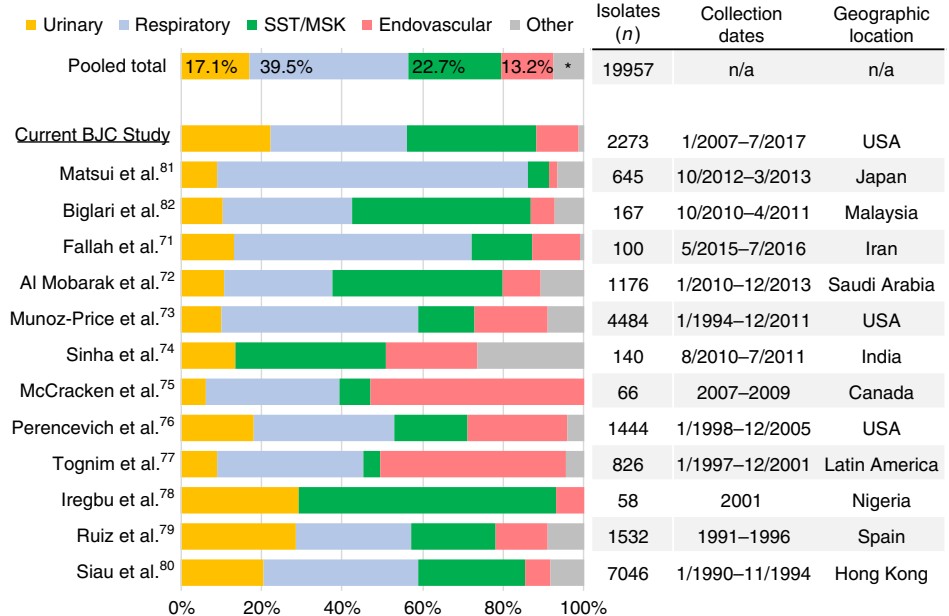

**Fig. 1** Urinary tract is an important source of *A. baumannii* clinical isolates. Graph depicts the distribution of isolates obtained from each anatomical site, listed by study and a "pooled total". Studies included in the analysis are listed on the left axis, with the current study underlined[71–82]. Percentage of pooled total isolates corresponding to each anatomical site, is noted in the top-most bar. Table along right axis lists the number of isolates included, the dates of isolate collection, and the country/region where each study was performed. *, "other" isolates comprise 7.5% of "Pooled Total" isolates. SST/MSK: soft tissue/musculoskeletal. Source data are provided as a Source Data file

Fg-coated catheters[21,22]. UPAB1 co-localized with aggregates of Fg (green fluorescence) (Fig. 2b, right panel), similar to what has been previously observed in corresponding models with *E. fae-calis* and MRSA[21,22]. Thus, UPAB1 infects the bladder and the catheter in a murine CAUTI model, in a similar manner to other uropathogens.

UPEC and MRSA are able to grow in healthy human urine in vitro[21,25] so we hypothesized that the inter-strain differences in the CAUTI model could be dictated by differential growth in urine. Indeed, 19606 displayed poorer growth and lower final bacterial concentrations in pooled healthy human urine, compared to UPAB1 (Fig. 2c), despite both strains displaying indistinguishable growth in rich media (Supplementary Fig. 1). Furthermore, UPAB1 grew better in pooled human urine when compared to established uropathogenic strains of UPEC and MRSA (Fig. 2c). Altogether, these results indicate that UPAB1 is a bona-fide uropathogenic *A. baumannii* (UPAB) strain.

Adhesion to catheters and urothelial cells is widely accepted as an essential step in establishing a CAUTI and/or UTI[26,27]. UPEC strains employ CUP pili to establish colonization in the bladder[25,28]. To validate the CAUTI model as a tool for identifying factors involved in *A. baumannii* uropathogenesis, we sequenced the 3.9 Mbp genome of UPAB1 (accession number CP032215-CP032220). We found that the UPAB1 genome harbors two loci encoding putative CUP pili, that we termed CUP1 and CUP2 (Fig. 3a). The CUP1 locus is homologous to the recently described type I CUP pilus locus, *prpABCD*[29], while the CUP2 locus has not been previously described. To assess whether these CUP pili contribute to UPAB1 uropathogenesis, we constructed the UPAB1 ΔCUP1,2 strain, which lacks the *ABCD* genes from CUP1 and CUP2 pili operons (Fig. 3a). Electron microscopy imaging of WT UPAB1 bacteria revealed distinct surface appendages, which were largely absent from the surfaces of ΔCUP1,2 bacteria (Fig. 3b). Despite both strains displaying indistinguishable growth in rich medium and healthy human urine (Supplementary Fig. 2), ΔCUP1,2 strains displayed a 2-log

reduction of bacterial burden on the implant and a 1-log reduction in the bladder, compared to WT UPAB1 in the CAUTI model (Fig. 3c). Altogether the CAUTI model was able to reveal that CUP pili, known adhesion factors for other bacterial uropathogens, are likely involved in establishing *A. baumannii* colonization of the urinary tract, validating the utility of the model to investigate *A. baumannii* uropathogenesis.

**Plasmid pAB5 dictates UPAB1 niche-specific virulence**. Having established the CAUTI model to evaluate *Acinetobacter* uropathogenesis, we analyzed UPAB1 for other potential virulence factors. Genome sequencing revealed UPAB1 contains two endogenous plasmids. One of these plasmids, named pAB5 (Accession numbers CP032216, CP032218 and CP032219) shares high homology with other plasmids belonging to the Large Conjugative Plasmid (LCP) family, pAB3 and pAB4[30,31]. LCPs contain a resistance island that encodes multiple putative antimicrobial resistance genes, and genes encoding the two TetR transcriptional regulators implicated in repression of the type VI secretion system (T6SS) in other *A. baumannii* strains[30]. Accordingly, a UPAB1 derivative lacking pAB5, UPAB1p-, displayed increased susceptibility to multiple antibiotics and activation of the T6SS, as observed by secretion of the T6SS-associated protein Hcp (Supplementary Fig. 3).

Repression of T6SS and/or other bacterial genes by pAB5 could impact the pathogenic potential of UPAB1. Alternatively, pAB5 could affect virulence by carrying virulence factors and/or pathogenicity islands, as has been extensively described in *Yersinia*, *Shigella*, and other bacterial pathogens[32,33]. Thus, we examined whether pAB5 influenced UPAB1 uropathogenesis. Wild-type UPAB1 (WT UPAB1) and UPAB1p- displayed comparable growth in vitro in both rich medium and urine from healthy donors (Supplementary Fig. 4), suggesting that carrying a large plasmid does not influence bacterial growth. In the CAUTI model, WT UPAB1 showed significantly higher bacterial burdens on the implant and in the bladder of the infected mice compared

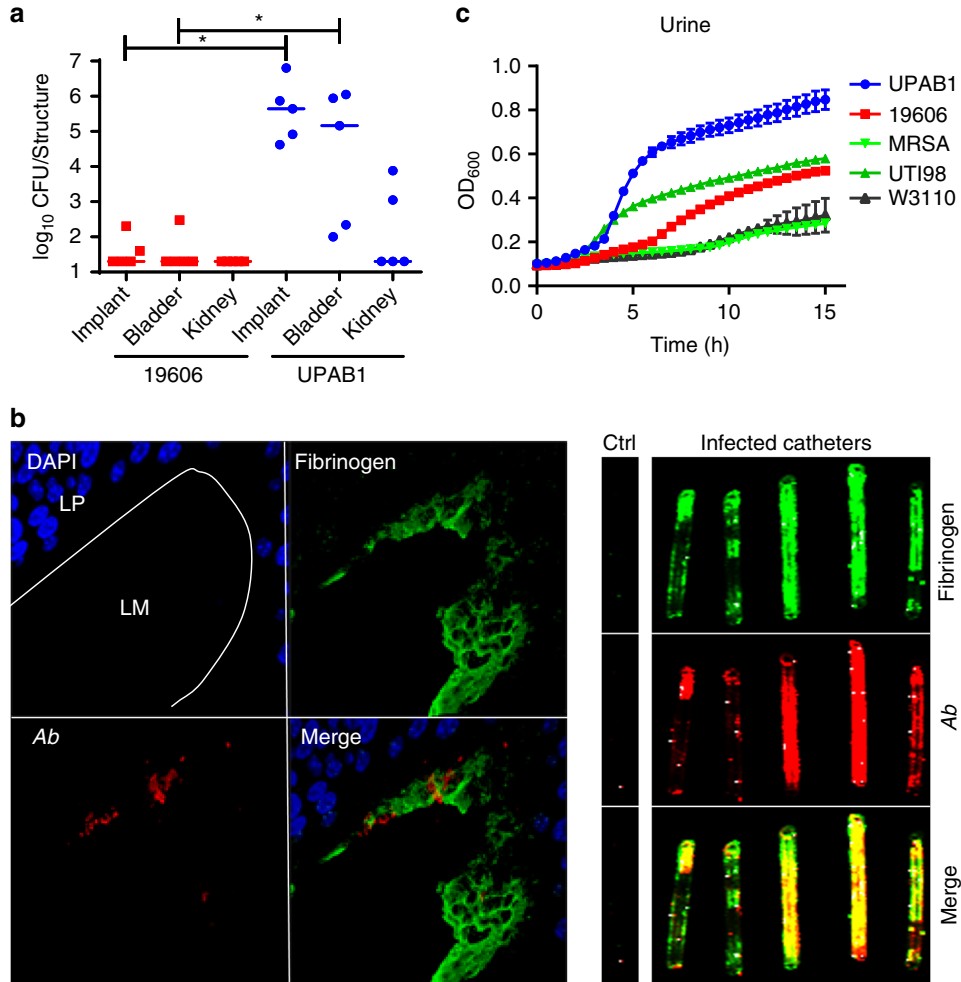

**Fig. 2** *A. baumannii* UPAB1 strain is an uropathogen. **a** Catheter-implanted mice were infected with ~$2 \times 10^8$ CFU of the indicated strains. Following 24 h of infection, total numbers of CFU recovered were determined for the implants, bladders and kidneys. Each symbol represents an individual mouse. For each strain, the median value is shown as the horizontal line. Statistical analyses were performed using the Mann–Whitney $U$ test, $*p < 0.05$. **b** Left panel: Representative images of an infected mouse bladder with implants at 24 hpi. Sections were stained for cell nuclei (blue), fibrinogen (green), and UPAB1 (red). Dotted white lines denote the separation of bladder tissue (LP: Lamina Propria) from lumen (LM). Bar: 20 µm. Right panel: UPAB1 colocalizes with fibrinogen deposited on implants at 24 hpi. **c** Growth curves of UPAB1, 19606, MRSA 1369, *E. coli* UTI89 and *E. coli* W3110 in healthy pooled urine as measured by OD600. The number of independent data points represented is four. Data represent mean and standard deviation values. Source data are provided as a Source Data file

to UPAB1p- (Fig. 4), demonstrating that the presence of pAB5 increases the virulence of *Acinetobacter* in CAUTI. We then tested if pAB5 carriage results in increased virulence in the acute pneumonia model, the most common model for evaluating *A. baumannii* pathogenesis[15]. Unexpectedly, 36 h after intranasal inoculation, we observed that mice infected with plasmid-containing UPAB1 WT carried one to four logs less bacterial burden in their lungs, spleens, livers, kidneys and hearts, compared to mice infected with UPAB1p- (Fig. 5). The differences in bacterial burdens were less pronounced in the lung than in the other organs, indicating that pAB5 effects are amplified during bacterial dissemination. Moreover, mice infected with UPAB1p- displayed 60% survival at 36 hpi, in contrast to 100% survival in mice infected with WT UPAB1. We appreciated no in vivo loss of pAB5, as determined by plating recovered bacteria on both LB agar and agar containing kanamycin and gentamycin, whose resistance cassettes are encoded on pAB5. In summary, pAB5 conferred improved UPAB1 colonization in the CAUTI model while attenuating the strain in a pneumonia model.

**pAB5 controls the expression of chromosomal factors in UPAB1**. Our results show that pAB5 confers UPAB1 niche-specific virulence. To identify whether potential effectors or secreted factors are differentially regulated by pAB5, we utilized a quantitative proteomics approach[34]. WT UPAB1 and UPAB1p- were grown in minimal medium to facilitate protein identification. As expected, several components and putative effectors of the T6SS were found in significantly lower levels in the presence of pAB5 (Fig. 6a and Supplementary Table 2). Additionally, levels of proteins involved in CUP1 and CUP2 pili assembly were reduced in WT UPAB1 (Fig. 6a), indicating that pAB5 can modulate expression of other chromosomally-encoded proteins, including some implicated in biofilm formation and adherence in other characterized uropathogens.

To identify additional UPAB1 factors differentially expressed in the presence of pAB5 we performed proteomic and transcriptomic experiments on whole bacterial cells. Since bacterial inocula used in murine experiments were prepared under shaking conditions (SHc) for the pneumonia model, and

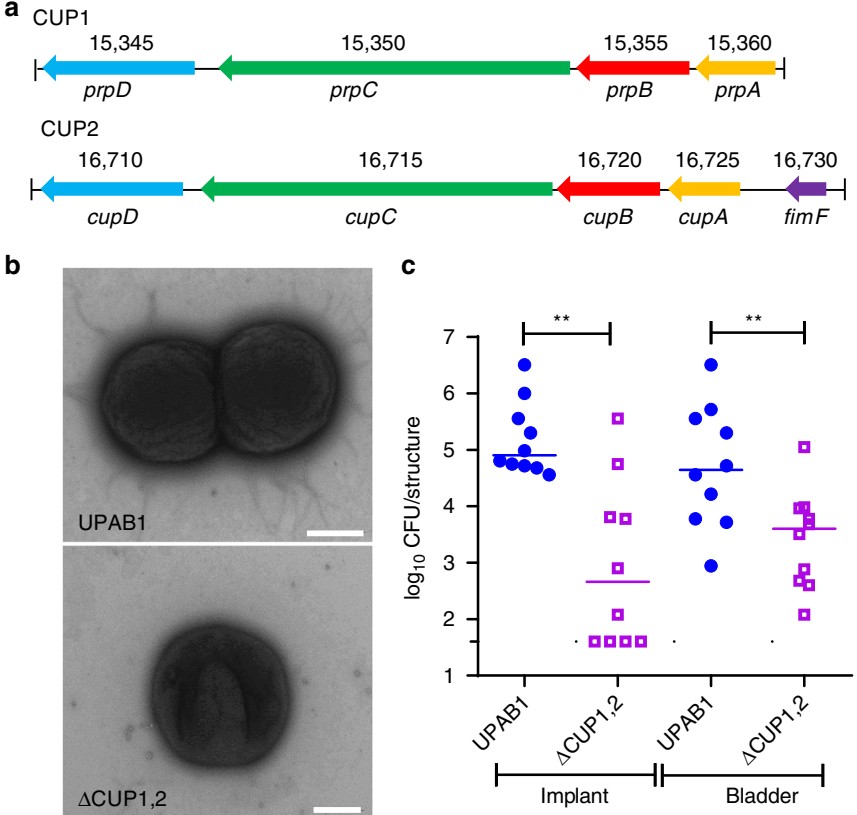

**Fig. 3** UPAB1 requires CUP pili to establish CAUTI. **a** Genetic organization of the CUP1 and CUP2 pili locus. **b** Transmission electron microscopic images showing the presence of pili structures in UPAB1 grown 2 by 24 h in static conditions. The pili-like structures were mostly absent in the ΔCUP1,2 mutant strain. Bars: 500 nm. **c** The ΔCUP1,2 mutant strain is impaired in the establishment of CAUTI. Each symbol represents an individual mouse. Data shown are pooled from two independent experiments. For each strain, the median value is shown as the horizontal line. Statistical analyses were performed using the Mann–Whitney $U$ test, $**p < 0.005$. Source data are provided as a Source Data file

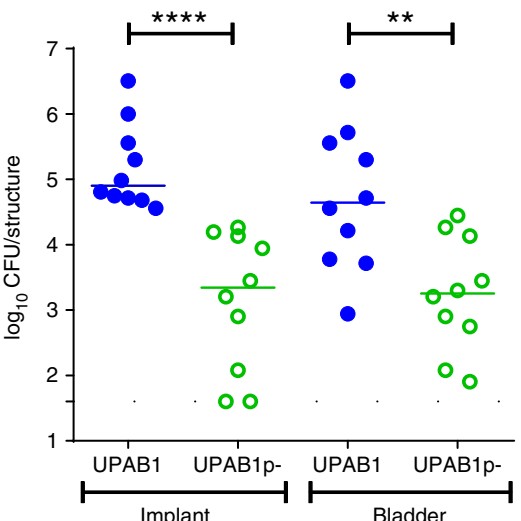

**Fig. 4** pAB5 is required for the establishment of CAUTI. Each symbol represents an individual mouse. Data shown are pooled from two independent experiments with five mice. For each strain, the median value is shown as the horizontal line. Statistical analyses were performed using the Mann–Whitney $U$ test, $**p < 0.005$, $****p < 0.0001$. Source data are provided as a Source Data file

static conditions (STc) for the CAUTI model, this analysis included bacteria grown in both conditions. Differential proteomic results are shown in Fig. 6b, c and RNAseq results are shown in Fig. 7a, b, respectively (Supplementary Table 3 and 4, Supplementary Data 1–5 and Supplementary Figs. 5 and 6). Proteomic and transcriptional analyses confirmed that pAB5 repressed T6SS in all the tested conditions. In SHc the expression of components of the CUP2 pili and genes involved in poly-N-acetyl-β-(1-6)-glucosamine (PNAG) biosynthesis were reduced by pAB5. PNAG is a surface polysaccharide that plays an important role in biofilm formation in vitro[35]. In addition, both proteomic and RNAseq experiments showed that pAB5 increased the expression of genes related to glutamate/aspartate transport[36] and the outer membrane protein OmpW (D1G37_01455) (Figs. 6b and 7a). OmpW is involved in iron uptake and colistin binding in *A. baumannii* 19606 and ATCC17978 strains[36]. In STc, CUP2 and PNAG synthesis were not repressed by pAB5. However, in these conditions, pAB5 repressed CUP1 pili and increased levels of putative glutamate/aspartate transport proteins, a T1SS secreted agglutinin RTX toxin (D1G37_00080), and an adjacently-encoded protein OmpA (D1G37_00085). The RTX toxin is homologous to the recently described putative RTX toxin in *A. nosocomialis* M2[34]. Overall, there was agreement between our proteomic and transcriptomic results. Our data indicate that in all experimental conditions tested, T6SS is repressed by pAB5. However, expression of other factors such as PNAG synthesis, CUP1, and CUP2, are differentially impacted according to growth

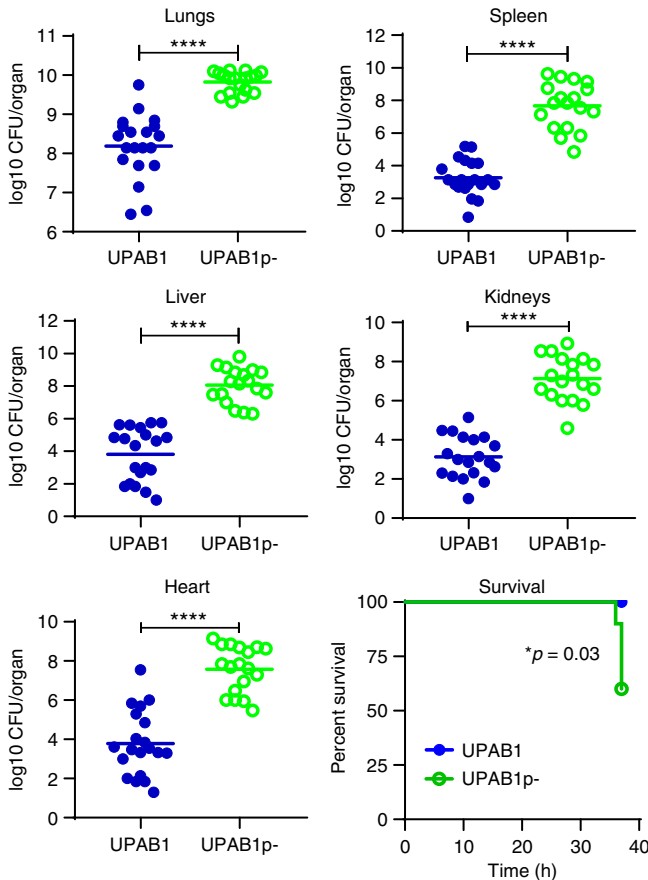

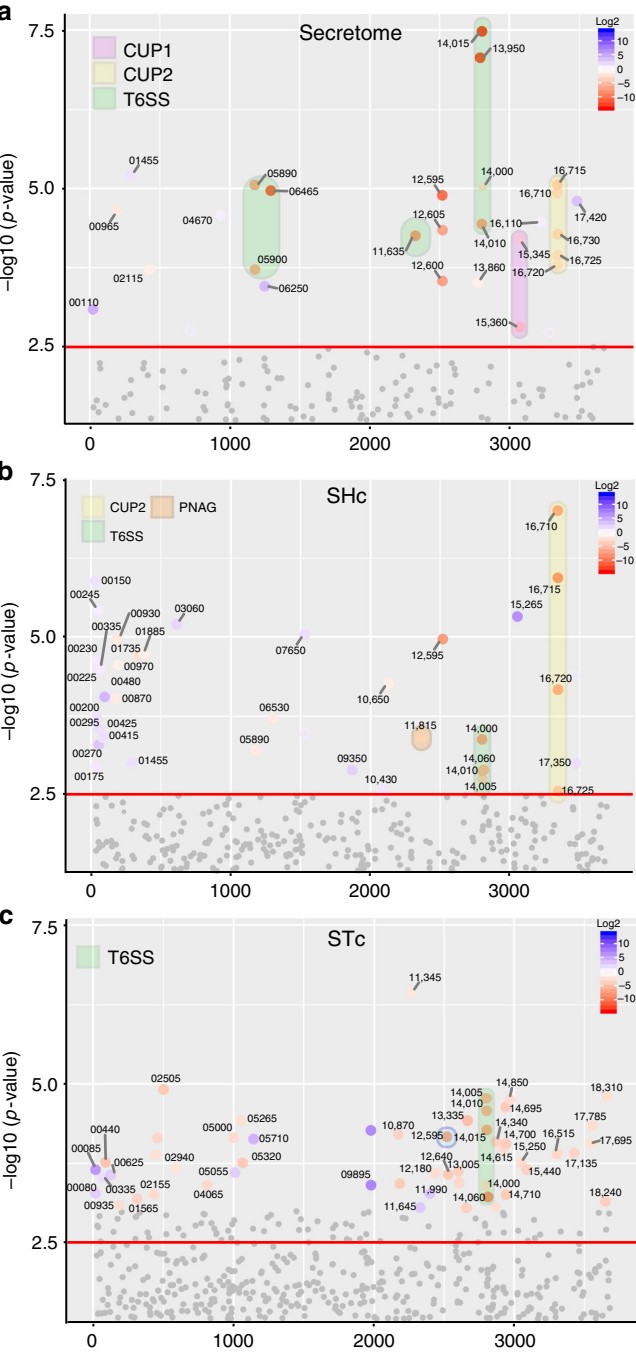

**Fig. 5** pAB5 is detrimental in an acute pneumonia murine model. Mice were intranasally inoculated with $1 \times 10^9$ CFU of either the WT UPAB1 or UPAB1p- strain. Following 36 h of infection, total numbers of CFU recovered were determined for lungs, spleen, liver, kidneys and heart. Each symbol represents an individual mouse. Data shown are pooled from two independent experiments. For each strain, the mean value is shown as the horizontal line. Statistical analyses were performed using the Mann–Whitney $U$ test, ****$p < 0.0001$. Source data are provided as a Source Data file

conditions. In conclusion, pAB5 modulates the expression of several chromosomally-encoded factors that could account for the differential outcomes in the CAUTI and pneumonia models.

**pAB5, but not other LCPs, impact PNAG formation in UPAB1.** PNAG is a surface polysaccharide that is a key virulence factor in highly divergent bacterial species[37,38]. *A. baumannii* PNAG plays a crucial role in biofilm formation[35]. Congo red dye associates with PNAG and is commonly employed to compare PNAG production in *A. baumannii*[35,39]. UPAB1p- grown on Congo red plates showed a clear production of PNAG (red colonies) while almost no production was observed in WT UPAB1 (white colonies) (Fig. 8a), validating our transcriptomic and proteomic data. Plasmids pAB3 and pAB4[30] are homologous to pAB5 and share common features, such as a conserved conjugation machinery and a regulatory locus containing the two TetR transcriptional regulators implicated in repressing T6SS[30]. However, the plasmids diverge in several regions, and additional genes putatively encoding transcriptional regulators are present on pAB5 (Supplementary Fig. 8). To examine whether the TetR repressors are also responsible for the pAB5-dependent phenotypes, we conjugated pAB3 and pAB4 into UPAB1. Although both plasmids repressed the T6SS in UPAB1 (Fig. 8b), we

**Fig. 6** pAB5 modulates chromosomally-encoded proteins. Quantitative proteome analysis of the effect of pAB5 on the secretome (**a**), the whole proteome with shaking (**b**), and whole proteome under static conditions (**c**). Manhattan plots demonstrating the significance of protein alteration, –log10(*p*-value), vs position in the genome are shown. The direction of the protein alteration is colored coded according the provided heat map. Selected proteins and accession numbers are shown. Full data sets are shown in Supplementary Fig. 7. Pink shadow highlights the genes encoding for CUP1 pili, yellow shadow highlights the genes encoding for CUP2 pili, orange shadow highlights the genes of PNAG biosynthetic pathway and green shadow highlights the genes encoding for the T6SS. Four biological replicates of each condition were analyzed. The default Maxquant FDR setting of 1% FDR at the protein and peptide levels were used

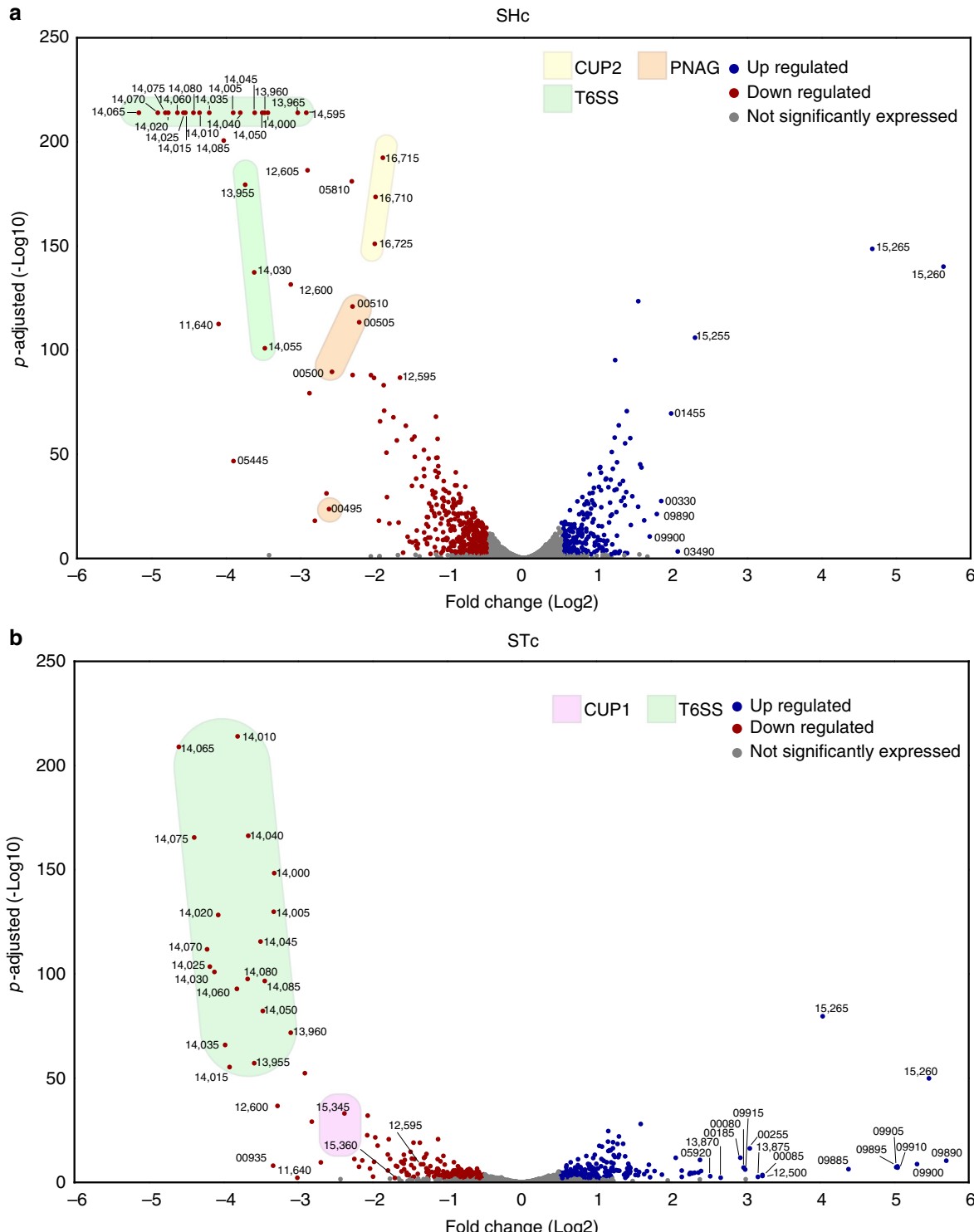

**Fig. 7** pAB5 differentially modifies the expression of chromosomal genes. DEGs with adjusted *p*-value < 0.1 for shaking (**a**) and static (**b**) grown conditions are depicted as a volcano plot to show the relationship between Log2 fold change and statistical significance. Each circle depicts one gene, upregulated genes are blue and downregulated genes are red, with the accession number shown for select genes. Pink shadow highlights the genes encoding for CUP1 pili, yellow shadow highlights the genes encoding for CUP2 pili, orange shadow highlights the genes of PNAG biosynthetic pathway and green shadow highlights the genes encoding for the T6SS. Three biological replicates of each condition were analyzed. All genes in the transcriptomic volcano plot with adjusted *p*-values above the statistical cut off of 0.1 (i.e., not significantly different between conditions) per the DESeq2 vignette are in gray, while the genes with adjusted *p*-values below 0.1 are blue or red

**a**

UPAB1

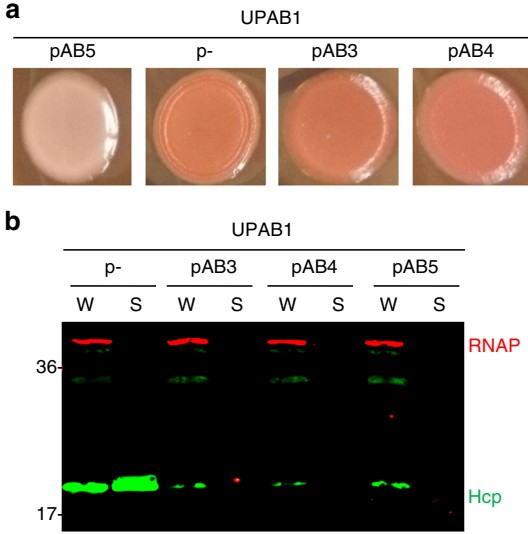

**b**

UPAB1

**Fig. 8** Effects of different LCPs in UPAB1. **a** Congo red agar plates, showing differences in the colony color (red versus white) associated with production of PNAG. **b** Western blot assays probing for Hcp (green) expression and secretion in whole-cell (W) or supernatants (S). RNA polymerase (RNAP, red) was used as a loading and lysis control. Source data are provided as a Source Data file

observed no significant repression of PNAG production in bacteria conjugated with pAB3 and pAB4 (Fig. 8a). Thus, repression of PNAG production is not result of TetR expression or fitness costs caused by plasmid carriage, indicating that additional features specific to pAB5 mediate chromosomal gene repression.

## Discussion

*Acinetobacter* infections are a threat to global health, especially due to the increasing frequency of MDR infections. Research on *Acinetobacter* pathogenesis has almost exclusively focused on pneumonia and bloodstream infections. In this study, we highlight the role of *A. baumannii* as a uropathogen. One in five *A. baumannii* clinical cases are associated with isolation from urinary sites, demonstrating the ability of this pathogen to persist in the human urinary tract. We describe strain-specific differences in the ability of *A. baumannii* to replicate in human urine and to establish colonization of catheterized murine bladders, with pathophysiology features akin to other well-established bacterial uropathogens. We established UPAB1 as a model strain for investigating *A. baumannii* uropathogenesis and identified CUP pili as probable determinants for bladder and catheter colonization. This model could be very useful for testing the ability of pilicides to reduce or prevent UTIs by MDR *A. baumannii* strains. Pilicides are small molecules with a 2 pyridone backbone that have been shown to inhibit type 1 and P pilus CUP assembly[40]. Future work will focus on determining the contribution of each individual CUP pilus in *A. baumannii* CAUTI.

UPAB1 opens new avenues for studying an important dimension of *Acinetobacter* disease. This extensively drug-resistant strain was isolated from a community-onset UTI in an Argentinian patient with no prior interaction with local healthcare systems. The strain belongs to an increasingly important *A. baumannii* phylogeny, multi-locus sequence type (ST) 25 (Pasteur scheme[41])/international clone (IC) 7. ST25/IC7 is one of the most prevalent clonal lineages associated with carbapenem-resistance and is associated with epidemic, endemic and sporadic strains isolated in multiple continents, including the Americas[42].

Though ST25/IC7 strains have been isolated from multiple infection sites, one study strongly linked ST25/IC7 to UTIs in domesticated animals[43]. Multiple ST25 strains harbor LCPs similar to pAB5[44], but LCPs have also been identified in *A. baumannii* urinary isolates belonging to other major international lineages[45]. Thus, investigations performed on UPAB1 are applicable to a relevant portion of *Acinetobacter* isolates, and a validated murine CAUTI model will be an important tool for developing interventions against the growing tide of MDR UTIs.

Remarkably, we found that the LCP pAB5 has a global influence on the physiology of UPAB1. pAB5 confers UPAB1 with improved survival in the urinary tract, but attenuates its virulence in a pulmonary model, highlighting the relevance of an under-appreciated feature of bacterial plasmids. LCPs like pAB5 were previously shown to repress T6SS in its bacterial host[30]. However, it was unclear whether LCPs mediate additional effects on gene expression. Here, we identified multiple surface molecules and metabolic pathways whose expression was affected by the presence of the plasmid, and these regulatory networks could explain the differential findings in murine models. In addition to the TetR regulators shown to be sufficient for T6SS suppression[30], pAB5 contains several putative transcriptional factors (Supplementary Fig. 8) that could directly alter the expression of chromosomally-encoded factors identified in our study. Alternatively, a plasmid-encoded factor might interact with unidentified, chromosomal master regulators that in turn impact the bacterial transcriptome. A third possibility is that presence of the plasmid is intrinsically sensed by the bacterium, triggering changes in the expression of several chromosomal factors. This latter scenario is not consistent with the observation that conjugation of pAB3 and pAB4 into UPAB1 does not alter PNAG expression. The fact that growth conditions also influence expression of virulence factors modulated by pAB5, indicates that pAB5 transcriptional effects are integrated into the global regulatory network of UPAB1. Virulence plasmids, i.e. plasmids encoding virulence factors, have been well-studied[32,33]. Although we cannot exclude the role of a putative plasmid-encoded virulence factor in CAUTI, our results show that pAB5 modulates the expression of multiple chromosomally-encoded factors, effecting the UPAB1 bacterial surface. Most *Acinetobacter* strains possess a constitutively active T6SS. LCP conjugation requires the survival of the recipient, which is the target of the T6SS. We have recently showed that T6SS repression by LCP enable plasmid dissemination and the propagation of MDR among *Acinetobacter* isolates[31]. Thus, while the ability to conjugate may be the driving force behind T6SS repression by LCP, whether regulation of other chromosomal factors is beneficial for the plasmid remains to be elucidated.

Bacterial plasmids that control chromosomally-encoded genes have been reported[46,47]. However, the broad impact on bacterial genome expression and pathobiology by pAB5 has not been reported for any other plasmid. Future work to elucidate the mechanisms through which pAB5 affects UPAB1 uropathogenesis will be needed. Since plasmid biology was largely researched prior to the "omics" era, revisiting these topics with modern tools may unveil additional effects that plasmids have on the expression of virulence factors of other bacteria. Even though the mechanistic details remain unclear, LCP carriage dictates differential *A. baumannii* survival in different host anatomical sites. Our findings challenge the dogma that opportunistic pathogens, such as *A. baumannii*, are niche nonspecific, and future investigations into the pathobiology of these bacteria must consider the clinical context of model strains.

## Methods

**Bacterial strains and growth conditions**. Bacterial strains used in this study are listed in Supplementary Table 5. Unless otherwise noted, strains were grown in

lysogeny broth (LB) liquid medium at 37 °C with shaking (200 rpm). The antibiotics rifampicin (100 μg per ml), kanamycin (7.5 or 15 μg per ml), hygromycin B (300 μg per ml), chloramphenicol (12.5 μg per ml), sulfamethoxazole (30 μg per ml), trimethoprim (6 μg per ml) and gentamicin (15 μg per ml) were added when necessary. Spontaneous rifampicin-resistant mutant strains were obtained by plating an overnight culture on LB agar with rifampicin.

**Local epidemiological analysis**. A retrospective study was performed on a cohort of cases identified in the BJC Healthcare System (BJC) from January 2007 to September 2017. Details on this analysis are published elsewhere[48]. Briefly, all cases with isolates obtained as a part of regular medical care and identified as "*Acinetobacter baumannii*" or "*Acinetobacter calcoaceticus-baumannii complex*" during the study period, were eligible for inclusion in this study. Both labels were used to account for differences in reporting methods implemented in the BJC over the course of this study. Cases identified solely on surveillance cultures during suspected outbreaks were excluded from the study, and only the first non-surveillance isolate per patient was included in the analysis. Cases were classified in one of five categories according to the anatomical site of isolation: "respiratory", "skin and soft tissue/musculoskeletal" (SST/MSK), "urinary", "endovascular", or "other."

**Systematic literature review of *Ab* epidemiology**. We performed a literature search in PubMed using combination of terms "*Acinetobacter baumannii*", "*Acinetobacter*", "epidemiology", "isolation sites," and an expanded search using bibliographies of identified studies. Reports published in or after 1995 were then reviewed in detail for inclusion criteria. For inclusion in our analysis, an epidemiological study must have fulfilled the following criteria: (1) reported isolates as *Acinetobacter baumannii* or *Acinetobacter calcoaceticus-baumannii complex*, i.e., studies that only characterized clinical isolates identified at the genus level alone were excluded; (2) performed on consecutive, non-duplicate isolates obtained from a geographically and temporally associated population; (3) isolates were not exclusively obtained from a single patient population (e.g., immunocompromised) or hospital ward (e.g., only ICU patients); and (4) included isolates obtained from "urinary" sites, and at least two other anatomical sites. Isolates in each study were then grouped into the five categories, as above, and enumerated.

**Construction of *A. baumannii* mutant strains**. An UPAB1 derivative that spontaneously loss pAB5 was identified by detection of restored Hcp secretion[49]. Briefly, individual colonies were inoculated in 96-well plates containing LB media and incubated overnight. Cultures were centrifuged, and 75 μL of supernatants were transferred into high-binding ELISA 96-well plates containing 25 μL binding buffer (100 mM sodium bicarbonate/carbonate, pH 9.6). Following overnight incubation at 4 °C, plates were washed with PBS, blocked, and then probed with rabbit anti-Hcp antibody. Bound anti-HCP antibodies were detected with HRP-conjugated goat anti-rabbit antibodies (Bio-Rad) and treatment with 100 μL of 3,3′,5,5′-Tetramethylbenzidine substrate (Cell Signaling Technologies). Cultures corresponding to wells where Hcp was present were subcloned for individual colonies with restored Hcp secretion. Loss of pAB5 was confirmed by loss aminoglycoside resistance and PCR with pAB5-specific primers. The primers used in this study are listed in Supplementary Table 6. Mutants were constructed as described previously[50]. Briefly, an antibiotic resistance cassette was amplified with 150 bp oligonucleotide primers (Integrated DNA Technologies) with homology to the flanking regions of the targeted gene with an additional 3′ 18–25 nucleotides of homology to the FRT site-flanked kanamycin resistance cassette from plasmid pKD4[51]. This PCR product was electroporated into competent UPAB1p- carrying pAT04, which expresses the RecAB recombinase[50]. Mutants were selected on 7.5 μg per ml kanamycin, and integration of the resistance marker was confirmed by PCR. To remove the kanamycin resistance cassette, electrocompetent mutants were transformed with pAT03 plasmid, which expresses the FLP recombinase. Spontaneous rifampicin-resistant clones of clean deletion mutants were obtained. Subsequently, pAB5 restoration in mutant strains was achieved by conjugation using UPAB1 wild type (WT) as donor strain and selection on LB agar containing gentamicin and rifampicin. Similar conjugation methods were employed to obtain UPAB1p- derivatives containing pAB3 and pAB4 from donor *A. baumannii* strains 17978 and AB04, respectively. All mutant strains were confirmed by antibiotic resistance profile, PCR, sequencing, and Hcp secretion assay (see below).

**Murine Model of *A. baumannii* CAUTI**. Six- to 8-wk-old female C57BL/6 mice were obtained from Charles River Laboratories. Mice were transurethrally implanted with a small piece of silicone tubing (catheter implant) and inoculated[52]. Briefly, mice were anesthetized by inhalation of 4% isoflurane, and a 4- to 5-mm piece of silicone tubing (catheter) was placed in the bladder via transurethral insertion. Bacterial strains were prepared for inoculation after static growth twice at 37 °C by centrifugation at 6500 rpm for 5 min, washing twice in 1 × PBS, and resuspension in PBS to the final inoculum. When indicated, mice were infected immediately following implant placement with ~2 × 10^8 CFUs bacteria in 50 μL via transurethral inoculation. At 24 h post-infection (pi), mice were euthanized, and kidneys, bladders, and implants were aseptically removed. The bacterial load present in each tissue was determined by homogenizing each organ in PBS and plating serial dilutions on LB agar supplemented with antibiotics when appropriate.

All CAUTI studies were performed in accordance with the guidelines of the Committee for Animal Studies at Washington University School of Medicine and we have complied with all relevant ethical regulations. Mice were housed with a cycle consisting of 12 h of light and dark with access to standard food and water ad libitum.

**Immunohistochemistry and Histopathology**. Bladder sections were deparaffinized with xylene (two times for 10 min), rehydrated with isopropanol (three times for 5 min), and washed with water for 5 min. Bladder antigens were exposed by boiling the section in 10 mM Na-citrate for 45 min and washed in water for 5 min and then PBS three times for 5 min each. Sections were then blocked with blocking buffer (PBS containing 1% BSA and 0.3% Triton X-100) for 1 h, washed with PBS containing 0.05% Tween 20 (PBS-T), and incubated with primary antibodies (1:100) overnight at 4 °C. Next, sections were washed three times with PBS, incubated with secondary antibodies (1:500, Alexa Fluor 488-labeled donkey anti-goat #A-11055 and Alexa Fluor 647-labeled donkey anti-goat #A-214473) for 1 h at room temperature, and washed three times with PBS. Lastly, sections were counterstained with Hoechst dye specific for DNA (1:20,000). A Zeiss Axioskop 2 MOT Plus microscope was used to analyze the sections via epifluorescence microscopy.

Implants were blocked with blocking buffer (1.5% BSA containing 0.1% sodium azide in PBS) overnight at 4 °C. Implants were then washed with PBS-T (three times for 5 min) and incubated with goat anti-fibrinogen and rabbit anti-UPAB1 (1:500) in dilution buffer (PBS containing 0.05% Tween 20, 0.1% BSA, and 0.5% methyl-α-D-mannopyranoside) at room temperature for 2 h. Implants were washed with PBS-T (three times for 5 min) and incubated with donkey anti-goat IRDye 800CW (#926-32214) and donkey anti-rabbit IRDye 680LT (#926-68023) (diluted 1:10,000) in dilution buffer for 1 h at room temperature. Lastly, implants were washed with PBS-T (three times for 5 min) and allowed to air dry. The Odyssey Imaging System (LI-COR Biosciences) was used to examine the infrared signal. Controls for autofluorescence included nonimplanted catheters.

**Murine model of *A. baumannii* acute pneumonia**. All infection experiments were approved by the Vanderbilt University Institutional Animal Care and Use Committee and we have complied with all relevant ethical regulations. Mice were housed with a 12 h/12 h light/dark cycle with access to standard food and water ad libitum. Lung infection experiments were performed as previously described[53]. Briefly, prior to infection, overnight cultures were diluted 1:1000 into 50 mL LB liquid media and incubated in 250-mL flasks at 37 °C under shaking conditions. Bacteria in logarithmic growth were harvested by centrifugation, washed twice with PBS, and resuspended in PBS to final inoculum. 9-week-old male C57BL/6 mice (Jackson Laboratories) were inoculated intranasally with a 7–8 × 10^8 CFUs of WT UPAB1 or UPAB1p- in 30 μL PBS. Lungs, livers, spleens, kidneys and hearts were aseptically harvested from mice euthanized at 36 h post-infection. Bacterial load was determined by homogenization of each organ and plating serial dilutions on LB agar.

**Hcp Western blotting**. Hcp western blots were performed as previously described[34]. Briefly, supernatants and whole cell samples were obtained from mid-log bacterial cultures. Samples were resuspended in 1X Laemmli buffer and loaded in a 12% polyacrylamide gel for separation. The mouse anti-Hcp[54] and rabbit anti-RNA polymerase (#663104, Biolegend, San Diego, CA) were both used at a concentration of 1:1000. Secondary IR dye antibodies from Licor were used at 1:10,000 (IRDye 800 CW goat anti-rabbit 926-32211 and IRDye 680 RD goat anti-mouse 925-68070). All blots were blocked in TBS blocking buffer (Licor).

**Genome sequence and annotation**. The DNA library of genomic DNA extracted from UPAB1 (DNeasy Blood and Tissue Kit, Qiagen) was prepared following the Pacific Biosciences 20 kb Template Preparation Using BluePippin Size-Selection System protocol. We sheared 7.5 μg of high-molecular-weight genomic DNA (final volume of 100 μL) using the Covaris g-TUBES (Covaris Inc.) at 4500 rpm (1900 × g) for 60 s on each side on an Eppendorf centrifuge 5424 (Eppendorf). The sheared DNA was size selected on a BluePippin system (Sage Science Inc.) using a cutoff range of 7 kb to 50 kb. The DNA damage repair, end repair, and SMRTbell ligation steps were performed as described in the template preparation protocol with the SMRTbell Template Prep Kit 1.0 reagents (Pacific Biosciences). The sequencing primer was annealed at a final concentration of 0.8333 nM, and the P4 polymerase was bound at 0.500 nM. The library was sequenced on a PacBio RSII instrument at a loading concentration (on-plate) of 80 pM using the MagBead loading protocol, DNA sequencing kit 2.0, SMRT cells v3, and 3-h movies. Sequencing was performed at the McGill University and Genome Quebec Innovation Center. Genome annotation was conducted using the NCBI Prokaryotic Genome Annotation Pipeline[55,56].

**Collection of urine**. Human urine was collected and pooled from three healthy female donors between 20–35 years of age. Donors had no history of kidney disease, diabetes or recent antibiotic treatment. Urine was sterilized using a 0.22 μm filter (EMD, Milipore) and adjusted to pH 6.0–6.5 prior to use. All participants have signed an informed consent and protocols were approved by the local Internal Review Board.

**Growth assays**. Bacteria were cultured overnight in rich liquid media (LB or brain-heart infusion, BHI) at 37 °C under shaking conditions. Cultures were washed with PBS and diluted to OD600 = 0.01 in 150 µL of either pooled human urine or rich media in 96 well plates and incubated at 37 °C under shaking conditions. OD600 values were measured every 30 min for 16 h via a BioTek microplate spectrophotometer. At least three separate experiments were performed with three wells per experiment for each strain.

**Transmission electron microscopy**. For negative staining and analysis by transmission electron microscopy, bacterial suspensions were allowed to absorb for 10 min onto freshly glow-discharged Formvar/carbon-coated copper grids. The grids were washed in distilled water and stained for 1 min with 1% aqueous uranyl acetate (Ted Pella Inc., Redding, CA). Excess liquid was gently removed, and grids were air dried. Samples were viewed on a JEOL 1200EX transmission electron microscope (JEOL USA, Peabody, MA) equipped with an AMT 8-megapixel digital camera (Advanced Microscopy Techniques, Woburn, MA).

**Secreted protein enrichment for quantitative proteomics analysis**. UPAB1 and UPAB1p- were grown overnight in 5 ml of M9 minimal medium supplemented with 0.2% casamino acids (M9CA) and subsequently diluted to A$_{600}$ = 0.05 in 50 ml of fresh M9CA. Cultures were incubated to mid-log phase and centrifuged at 15,000 g for 2 min. Supernatants were filter-sterilized with Steriflip vacuum-driven filtration devices (Millipore), concentrated with an Amicon Ultra (Millipore) concentrator with a 10-kDa molecular weight cutoff, flash frozen and lyophilized. Lyophilized samples were then processed for mass spectrometry analysis as described below. Four individual 50-ml culture biological replicates were prepared for each strain.

**Whole cells samples for transcriptomic and proteomics analysis**. Samples were prepared in the same conditions as the inocula used for the murine pneumonia model experiments (shaking conditions, SHc), and for the CAUTI model (static conditions, STc). For SHc samples, overnight cultures were diluted 1:1000 into 50 mL LB liquid media and incubated in 250-mL flasks at 37 °C under shaking conditions. Bacteria in logarithmic growth were harvested by centrifugation. For STc samples, overnight cultures grown under static conditions were diluted 1:200 into 15 mL LB liquid media and incubated in 50-mL flasks at 37 °C under static conditions. Overnight cultures were harvested by centrifugation. Samples for transcriptomics were resuspended in (RNAprotect Cell Reagent (Quiagen) and processed as described below. Samples for comparative proteomic analysis were flash frozen and lyophilized. Lyophilized samples were then processed for mass spectrometry analysis as described below

**Preparation of protein samples for proteomic analysis**. Lyophilized supernatant samples (containing secreted proteins) or lyophilized bacterial pellets were resuspended in 4% SDS, 100 mM HEPES, 10 mM DTT and boiled at 95 °C for 10 min with shaking at 2000 rpm to solubilize protein material. 100 µg of protein, as determined by BCA protein quantification assay (Thermo Fisher Scientific), was precipitated by overnight incubation in 80% acetone at −20 °C with shaking. Precipitated proteins were centrifuged at 10,000 G for 10 mins at 0 °C. To ensure removal of SDS, resulting pellets were again suspended and incubated overnight in 80% acetone. Precipitated protein pellets were obtained by centrifugation and dried at 75 °C for 5 mins and stored at −20 °C until analyzed.

**Digestion of complex protein lysates**. Hundred micrograms of dried protein pellets were prepared[57] with minor alterations. Briefly, dried protein samples were resuspended in 6 M urea, 2 M thiourea, 40 mM NH$_4$HCO$_3$ and reduced/alkylated with TCEP/Chloroacetamide (final concentration 10 mM and 20 mM respectively) for 1 h in the dark. Samples were pre-digested with Lys-C (1/200 w/w) for 3 h then diluted to a final urea/thiourea concentration below 2 M and digested overnight with trypsin (1/50 w/w). Digested samples were acidified to a final concentration of 0.5% formic acid and desalted with home-made high-capacity StageTips composed on 5 mg Empore™ C18 material (3 M, Maplewood, Minnesota) and 5 mg of OLIGO R3 reverse phase resin (Thermo Fisher Scientific) according to the protocol Ishihama and Rappsilber[58,59]. Using this protocol StageTips were first conditions with 10 bed volumes of Buffer B (80% ACN, 0.1% FA) and then equilibrated with 10 bed volumes of Buffer A* (2% ACN, 0.01%TFA). Peptide samples were loaded onto columns and columns washed with 10 bed volumes of Buffer A*. Bound peptides were eluted with buffer B (80% ACN, 0.1% Formic acid), dried and stored at −20 °C.

**Reversed phase LC-MS**. Purified peptides were resuspended in Buffer A* (2% ACN, 0.01%TFA) and separated using a two-column chromatography set up comprising a PepMap100 C18 20 mm × 75 µm trap and a PepMap C18 500 mm × 75 µm analytical column (Thermo Scientific). Samples were concentrated onto the trap column at 5 µl/min with buffer A (2% ACN, 0.1% FA) for 5 mins and infused into either an Orbitrap Elite™ Mass Spectrometer (Thermo Scientific) or Q-Exactive™ plus Mass Spectrometer (Thermo Scientific) at 300 nl/min via the analytical column using Dionex Ultimate 3000 UPLCs (Thermo Scientific). For comparison

of the effect of the absence vs presence of pAB5, whole cell proteomes samples were analyzed on the Q-Exactive™ plus using an 125 min gradients altering the buffer composition from 1% buffer B to 28% B (80% ACN, 0.1% FA) over 95 mins, then from 28% B to 40% B over 10 mins, then from 40% B to 100% B over 2 mins, the composition was held at 100% B for 3 mins, and then dropped to 3% B over 5 mins and held at 3% B for another 10 mins. The Q-Exactive™ plus Mass Spectrometer was operated in a data-dependent mode automatically switching between the acquisition of a single Orbitrap MS scan (70,000 resolution) followed by 15 data-dependent HCD MS-MS events (resolution 35 k AGC target of $2 \times 10^5$ with a maximum injection time of 110 ms, NCE 35) were allowed with 45 s dynamic exclusion enabled. Four biological replicates of each whole proteome condition were analyzed in a randomized manner to minimize batch effects. Wash runs were performed between replicates to prevent samples carry over.

The analysis of the effect of the absence vs presence of pAB5 on the secretome and under differencing conditions was performed on an Orbitrap Elite™ using an 142 min gradient altering the buffer composition from 1% buffer B to 28% B over 115 min, then from 28% B to 40% B over 7 , then from 40% B to 100% B over 2 min, the composition was held at 100% B for 3 min, and then dropped to 3% B over 5 min and held at 3% B for another 10 min. The Elite™ Mass Spectrometer was operated in a data-dependent mode automatically switching between the acquisition of a single Orbitrap MS scan (60,000 resolution) followed by five data-dependent HCD MS-MS events (resolution 15 k AGC target of $4 \times 10^5$ with a maximum injection time of 200 ms, NCE 35) were allowed with 35 ss dynamic exclusion enabled. Four biological replicates of each whole proteome condition were analyzed in a randomized manner to minimize batch effects. Wash runs were performed between replicates to prevent samples carry over

**Proteomic data analysis**. MS data were processed using MaxQuant (v1.5.3.30[60]). Database searching was carried out against *Acinetobacter baumannii* UPAB1 (GenBank entry: CP032215, CP032216, CP032217, CP032218, CP032219, and CP032220). Searchers were undertaken with the following search parameters: carbamidomethylation of cysteine as a fixed modification; oxidation of methionine, acetylation of protein N-terminal trypsin/P cleavage with a maximum of two missed cleavages. The default Maxquant FDR setting of 1% FDR at the protein and peptide levels was used. To enhance the identification of peptides between samples, the Match between Runs option was enabled with a precursor match window set to 2 min and an alignment window of 10 min. For label free quantitation the MaxLFQ option within Maxquant[61] was enabled in addition to the re-quantification module. The resulting outputs were processed within the Perseus (v1.5.0.9)[62] analysis environment to remove reverse matches and common proteins contaminates prior to further analysis. For LFQ comparisons missing values were imputed with a downshift of 2.5 and width of 0.3 standard derivations. Statistically assessment of alterations between conditions was done using two sample *t*-test within Perseus with a Benjamini Hochberg correction FDR of 0.05. Pearson correlations and Perseus outputs were visualized using R (https://www.r-project.org/). The mass spectrometry proteomics data have been deposited to the ProteomeXchange Consortium via the PRIDE[63] partner repository with the dataset identifier PXD011302 and PXD011341 which contains all the raw MS data files and Maxquant outputs, including the list of all search parameters.

**RNA-sequencing extraction and sequencing**. Triplicate cultures of WT UPAB1 (UPAB1_pAB5) or UPAB1p- (UPAB1_plasmid-) were grown in LB broth, Lennox (Fisher Bio Reagents, Cat. No BP1427) to mid-log phase at 37 °C under either shaking or static conditions. Samples were added to RNAlater™ stabilization solution (Thermo Fisher Scientific, Waltham, MA, USA) and kept at −80 °C until processing. Total RNA was extracted from the thawed samples with the ZR Fungal/ Bacterial RNA MiniPrep™ kit (Zymo Research, Irvine, CA, USA) using the RNA protocol following manufactures instructions except with an additional on column TURBO DNase treatment (Thermo Fisher Scientific) for 30 min at 37 °C to remove genomic DNA. Following TURBO DNase treatment, the RNA was purified in the RNA Clean & Concentrator kit (Zymo Research). The RNA samples had concentration, absorbance values at 260/280 nm, and absorbance values 260/230 nm analyzed by NanoVue Plus™ Spectrophotometer (GE Healthcare, Chicago, IL, USA). Contaminating gDNA was analyzed using PCR amplification of the 16S rRNA gene. Approximately 2 µg of RNA was used as input for the Ribo-Zero rRNA Removal Kit (Illumina, San Diego, CA). The rRNA-free sample was converted into cDNA libraries using SuperScript II (ThermoFisher Scientific), DNA Pol I (New England Biolabs, Ipswich, MA, CA), and DNA ligase (New England Biolabs)[64]. cDNA was quantified using Qubit dsDNA HS assay (ThermoFisher Scientific). Five nanograms of DNA was used as input to make Illumina sequencing libraries with the Nextera XT Kit (Illumina)[65].

**Transcriptomic analysis**. Pooled cDNA libraries were submitted to the Center for Genome Sciences & Systems Biology at Washington University in St. Louis School of Medicine. Samples were sequenced on an Illumina NextSeq 550 system to obtain 1 × 75 bp sequences. Raw reads were demultiplexed by barcodes and had adapters removed with trimmomatic v.38using the command "java -Xms1024m -Xmx1024m -jar<trimmomatic_jar>SE -phred33 -trimlog<-trimlog_output><multiplexed_read><trimmed_read>ILLUMINACLIP:/opt/apps/

trimmomatic/0.36/adapters/NexteraPE-PE.fa:2:30:10 LEADING:3 TRAILING:3 SLIDINGWINDOW:4:15 MINLEN:36"[66]. The UPAB1 genome (Accession CP032215-20) was converted into a bowtie2 v.2.3.4.1 index with command "bowtie2-build UPAB1.fasta<index>" and the trimmed reads were aligned to it with command "bowtie2 -x<index>-U<trimmed_read>-S<sam_output>2><bowtie2_log>" to generate SAM files[67]. Count matrices were generated using FeatureCounts within subread v1.5.3 with command "srun featureCounts -a<SAF_file>-F SAF -o<count_output><sam_file>"[68]. Differential expression analysis of the count matrix was performed using DESeq2[69]. Per the DESeq2 vignette (http://bioconductor.org/packages/devel/bioc/vignettes/DESeq2/inst/doc/DESeq2.html), genes with counts <10 were discarded from differential expression analysis. Variance stabilizing transformation of read count was analyzed as a heatmap and principal component figure. DEG analysis was performed by comparing UPAB1 with pAB5 versus UPAB1 cured of pAB5 separately for shaking and static growth. Per the DESeq2 vignette, genes with adjusted p-values <.1 were determined to be significantly differentially expressed.

**PNAG detection by Congo Red plates**. The colony morphology of *A. baumannii* strains was studied on plates containing Congo red agar composed of SOBG media (SOB media+2% glycerol) supplemented with 40 μg/ml of Congo red (Sigma Chemical Co., St. Louis, MO)[35,70]. Plates were incubated at 28 °C for 24 h. On these plates PNAG-synthesizing cells produced red colonies, whereas PNAG-deficient cells produced white colonies.

**Statistical analysis**. All statistical analyses were performed using GraphPad Prism (GraphPad Software Inc., La Jolla, CA). For comparison of bacterial load in the murine CAUTI model, statistical analyses were performed using the Mann–Whitney $U$ test. For the murine acute pneumonia model, data were log transformed and analyzed for Gaussian distribution using the D'Angostino-Pearson omnibus normality test. Data sets displaying Gaussian distribution were then analyzed by one-way ANOVA with Tukey's test for multiple comparisons. Data sets displaying non-Gaussian distribution were analyzed by Kruskal-Wallis test with Dunn's test for multiple comparisons. For all other assays, normal data distribution was checked with the Shapiro-Wilk normality test. For normally distributed datasets, parametric one-way ANOVA was performed with Tukey's correction for multiple comparisons. For non-normally distributed datasets, nonparametric Kruskal-Wallis test with Dunn's correction for multiple comparisons was used.

**Reporting summary**. Further information on research design is available in the Nature Research Reporting Summary linked to this article.

## Data availability

The authors declare that data supporting the findings of this study are available within the paper and its supplementary files. The source data underlying Figs. 1, 2, 3b, c, 4, 5, 8 and Supplementary Figs. 1, 2, 3 and 4 are provided as a Source Data file. The whole-genome sequence project was deposited in the Sequence Read Archive (SRA) at the National Center for Biotechnology Information (NCBI) under the accession number PRJNA487603, and the whole-genome sequences were deposited in the GenBank database under the accession numbers CP032215 to CP032220. Processed RNA-seq reads have been submitted to the Short Read Archive under BioProject PRJNA499107.

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

## Acknowledgements

We thank the imaging laboratory of Molecular Microbiology Department at Washington University in St Louis, especially W. Beatty for EM analysis. We thank to the members of the Feldman lab for critical reading of the manuscript. This work was supported by grants from the National Institute of Allergy and Infectious Diseases (grants T32AI007171-38, R01AI144120 and R01DK051406).

## Author contributions

Conceived and designed the experiments: G.D.V., A.L.F.-M., J.J.C., L.D.P., G.D., E.P.S., S.J.H., and M.F.F. Performed the experiments: G.D.V., A.L.F.-M., J.J.C., N.E.S., R.F.P., L.D.P., M.E.H., and B.W. Analyzed the data: G.D.V., A.L.F.-M., J.J.C., M.F.H., N.E.S., R.F.P., L.D.P., G.D., E.P.S., S.J.H., and M.F.F. Isolated UPAB1: L.F. Wrote the paper: G.D.V., J.J.C., and M.F.F.

## Additional information

**Competing interests:** The authors declare no competing interests.

