## [Peer Review File · Nature Communications]

Reviewers' comments:

Reviewer #1 (Remarks to the Author):

This manuscript examined the role of an endogenous plasmid pAB5 in uropathogenic *Acinetobacter baumannii*. Please see my comments below which might help strengthen the manuscript.

1. It is a comprehensive study; however, considering the novelty, I am afraid that it is more suitable for a leading discipline journal.
2. Another major concern is that the bioinformatic analysis of the transcriptomics and proteomics results is preliminary. Comprehensive pathway analysis might help to generate some hypothesis on how pAB5 repressed the expression of the related chromosomal genes.
3. Please note *A. baumannii* ATCC 19606 was originally isolated from urine (http://www.lgcstandards-atcc.org/products/all/19606.aspx?geo_country=gb). Did the authors try to transform pAB5 into ATCC 19606 and then investigate the virulence and its ability to cause UTI?
4. Lines 262-263, which broth was used for the culture? The culture conditions can substantially affect the transcriptomics and proteomics results for both wild-type and mutants.
5. Lines 428, did the authors delete and complement the *pga* locus in UPAB1? Such results will confirm the definite role of PNAG in UTI.
6. Lines 457-458, clearly, it would be difficult to image that the repression of PNAG production was due to TetR expression. The feature(s) specific to pAB5-mediate chromosomal gene repression should be elucidated in this study, in particular those related to UTI.
7. Lines 493, had the authors considered knocking out the putative transcriptional factors to investigate their roles?
8. Figures 3-5, could the authors explain the large variability (~3-4 log₁₀ CFU/organ) in the bacterial burden results? In Figure 5, did the authors do colony PCR to confirm the exist of the plasmid pAB5?
9. Tables S4-S5, why not provide the fold change results for each differentially expressed protein?

Minor comments:

10. The Results section contains a number of sentences on the methods and can be more concise. The Discussion section is somewhat brief and superficial.

Reviewer #2 (Remarks to the Author):

In this study the authors present a novel, previously not observed mechanism in *Acinetobacter baumannii* isolates, enabling or enhancing colonisation in the urinary tract.

Comments:

Uropathogens exhibit specific mechanisms, which enable them to survive and grow in urine. *A. baumannii* up to now is a rarely reported isolate in urinary tract infections. The authors have performed a retrospective analysis of their databank revealing that approximately 20% of *A. baumannii* isolates are cultured from the urinary tract. The statement that *Acinetobacter baumannii* is a leading source of CAUTI however needs further proof. In most bacterial spectrum analyses of UTI or CAUTI *A. baumannii* is reported in less than 2% of the entire bacterial spectrum. It would therefore be interesting to see the figure of the authors epidemiological study. What was the percentage of *A. baumannii* of all urinary isolates identified in their study? The references cited mainly describe *A. baumannii* in specific situations such as patients in ICU. Do the authors have an explanation from their own work, what might be the driving force for this. Is it the high antibiotic resistance, enabling *A. baumannii* to be selected during antibiotic treatment in intensive care patients, or are there additional factors.

The isolated strain UPAB1 was grown in a patient from Argentina. Do the authors have information about the severity of the UTI of this patient? Was it urosepsis, or asymptomatic bacteriuria or something else? In this direction, do the authors have information about the different types of UTI found in their epidemiological study? This could answer the issue about pathogenicity of *A. baumannii* in UTI from a clinical epidemiological point of view.

The authors robustly describe the role of the plasmid pAB5. Do they have information about, in how many strains of *A. baumannii* plasmid pAB5 has been identified? In addition is the plasmid pAB5 genetically identical or comparable to plasmid found in other typical uropathogens such as *E. coli* or other enterobacteria.

F. Wagenlehner

Reviewer #3 (Remarks to the Author):

The manuscript by Venanzio et al. describes an interesting phenomenon, which is the regulation of chromosomal genes by the presence of a plasmid in *Acinetobacter baumannii*. Additionally, this regulation by the plasmids leads to a change in virulence and niche specificity. The manuscript is well structured, clearly written and concise. There are only a few things that should be clarified or extended upon.

1. The methods are in very many places greatly abbreviated and not fully described. In these cases previous literature is referenced for details. However, a lot of this literature is pay-walled. The methods should be described in enough detail to allow replication without having to go to non-open access journals. This applies to almost every method section. For example, in line 118 you mention the construction of the strain without pAB5, which is the most central piece of the methods in the whole manuscript, as all subsequent work depends on it. However, no details on strain construction are provided and the referenced publication is not open access.
2. Line 212 and following, the 100 ug dried protein pellets seem to come out of nowhere, there are no descriptions in the whole methods part on from what and how this protein was prepared.
3. Line 237, there is no description anywhere in the manuscript on how the secretome samples were prepared.
4. Line 253, more detail on the MaxQuant search parameters would be useful. At least FDRs for protein and peptide identification should be reported.
5. Both in the methods and the results there is no mention of the number of replicates used for proteomics and transcriptomics. Based on the proteomic raw data I am guessing that the replicate numbers were appropriate, but this needs to be clearly documented in the text.
6. FDRs and replicate numbers should also be reported in figure captions for proteomic and transcriptomic data.
7. The Type 6 secretion system should be mentioned in the abstract or in the keywords to increase visibility of this very important topic in search results
8. The abstract is very well written. One point I think could be made more clear is the fact that removal of the plasmid leads to increased infection in the respiratory tract as this is a concrete and critical result.

Signed: Manuel Kleiner

Reviewers' comments:

Reviewer #1 (Remarks to the Author):

This manuscript examined the role of an endogenous plasmid pAB5 in uropathogenic *Acinetobacter baumannii*. Please see my comments below which might help strengthen the manuscript.

1. It is a comprehensive study; however, considering the novelty, I am afraid that it is more suitable for a leading discipline journal.

We thank the reviewer for his comments and suggestions, which we will address here. We respectfully disagree with the reviewer's opinion regarding novelty. We believe this work is novel given the following points:

- 1) We developed and validated the first catheter-associated urinary tract infection (CAUTI) model to investigate *A. baumannii* uropathogenesis.
- 2) Employing this model, we show that UPAB1, a community-onset MDR UTI strain, requires CUP pili, including a previously undescribed pilus, to efficiently colonize the bladder and catheter. CUP pili have never been shown to be relevant in *Acinetobacter* infection.
- 3) Employing our model, we determined that presence of the large conjugative plasmid pAB5 greatly increases UPAB1 survival in a CAUTI model, but dramatically diminishes the bacterial burden in a pneumonia model. These findings challenge the dogma that opportunistic pathogens, such as *A. baumannii*, are niche unspecific.
- 4) We employed differential proteomic and transcriptomic experiments to show that the niche specificity conferred by pAB5 is due to the remarkable ability of this plasmid to control the expression of multiple chromosomally-encoded virulence factors in UPAB1. To our knowledge, the level of control of the chromosome exerted by the plasmid is unprecedented.
- 5) We report a retrospective epidemiological study and found that ~20% of *A. baumannii* clinical isolates, a bacterium commonly associated with respiratory infections and bacteremia, are obtained from urinary sources. This is not recognized in the current literature.

6) Our results demonstrate the importance of considering the clinical context of *A. baumannii* strains to understand *Acinetobacter* pathogenesis.

Taking these points into consideration, we believe our data will be of interest for the *Acinetobacter* research community, including clinicians, but also for anyone interested in microbiology and genetics.

2. Another major concern is that the bioinformatic analysis of the transcriptomics and proteomics results is preliminary. Comprehensive pathway analysis might help to generate some hypothesis on how pAB5 repressed the expression of the related chromosomal genes.

We have utilized the most sophisticated comparative transcriptomic and proteomic techniques available. The question of how pAB5 regulates chromosomally encoded factors is extremely interesting and it constitutes the whole aim 3 of my recently awarded R01 grant from NIH. We hope to obtain the answer to this question in the future.

3. Please note *A. baumannii* ATCC 19606 was originally isolated from urine (http://www.lgcstandards-atcc.org/products/all/19606.aspx?geo_country=gb). Did the authors try to transform pAB5 into ATCC 19606 and then investigate the virulence and its ability to cause UTI?

We thank the reviewer for this observation. We are aware that 19606 is a urinary isolate and that was included in the original submission (lines 112-113). This observation and the fact that 19606 has been broadly used as a model strain justify the inclusion of this strain in our studies.

We agree with the reviewer that introducing pAB5 into 19606 would be very interesting. Unfortunately, our efforts to introduce pAB5 into ATCC 19606 have not been successful. We extensively tried both transformation and conjugation strategies. As previously described (Multidrug-resistant plasmids repress chromosomally encoded T6SS to enable their dissemination. Di Venanzio et al, PNAS 2019), efficiency of conjugation is reduced by an active type VI secretion system and any other contact-dependent killing mechanism from the donor and the recipient strain, and ATCC19606 strain has an extremely active T6SS. Also, UPAB1 is MDR, which makes selection of transconjugants in conjugation assays difficult. Simultaneously, three different protocols for isolating large plasmids were used to try to purify pAB5. Although we were able to purify the plasmid, no transformants were obtained by electroporation or chemical transformation. We hope in the future to obtain additional plasmids and strains that will enable further combinations to extend our understanding of the effects of LCPs in other urinary isolates.

4. Lines 262-263, which broth was used for the culture? The culture conditions can substantially affect the transcriptomics and proteomics results for both wild-type and mutants.

We agree with the reviewer that the culture conditions can substantially affect the transcriptomic and proteomic results, in our and all studies. The information about

growth conditions (media, shaking vs static conditions, incubation times) was dispersed throughout the text. To further clarify, we included new sections in Materials and Methods (lines 415-432).

5. Lines 428, did the authors delete and complement the *pga* locus in UPAB1? Such results will confirm the definite role of PNAG in UTI.

We agree with the reviewer that such experiments would help to determine the role of PNAG in UTI. We cannot and have not made the claim that PNAG plays a role in uropathogenesis, but simply use it as a visual marker of the global effects of pAB5 in UPAB1 physiology. Elucidating the individual contribution of each single pAB5-regulated putative virulence factor in *Acinetobacter baumannii* UTI is beyond the scope of this current work but is the focus of Aim 2 in my recently awarded R01 grant.

6. Lines 457-458, clearly, it would be difficult to image that the repression of PNAG production was due to TetR expression. The feature(s) specific to pAB5-mediate chromosomal gene repression should be elucidated in this study, in particular those related to UTI.

pAB5 contains several regulators and in this manuscript, we do not claim which regulator acts on each gene. We agree with the reviewer that identifying the pAB5 features that control chromosomal gene expression, particularly those involved in the establishment of CAUTI, is important. However, understanding what regulator specifically represses each gene will not alter any of the conclusions of our present work. Answering that question requires extensive mutagenesis of MDR plasmids and is not straightforward. As mentioned before, investigating how pAB5 regulates chromosomally encoded factors constitutes the entirety of aim 3 of my recently awarded R01 grant from the NIH.

7. Lines 493, had the authors considered knocking out the putative transcriptional factors to investigate their roles?

The answer to comment 6 also applies here.

8. Figures 3-5, could the authors explain the large variability (~3-4 log₁₀ CFU/organ) in the bacterial burden results? In Figure 5, did the authors do colony PCR to confirm the exist of the plasmid pAB5?

We appreciate the reviewer's observations and agree that our data exhibit variability in bacterial burden. This degree of variability appears to be inherent to the model and is attributed to differences between individual mice. Our observed variability is comparable to the ones seen in prior work examining other pathogens in the same models (See: Xu, W. et al. Host and bacterial proteases influence biofilm formation and virulence in a murine model of enterococcal catheter-associated urinary tract infection. *npj Biofilms Microbiomes* 3, 28 (2017).

Colomer-Winter, C. et al. Manganese acquisition is essential for virulence of *Enterococcus faecalis*. (2018). doi:10.1371/journal.ppat.1007102.

Choby, J. E. et al. PheWAS uncovers a pathological role of coagulation Factor X during *Acinetobacter*. (2019). doi:10.1128/IAI.00031-19.

Loneragan, Z. R. et al. An *Acinetobacter baumannii*, Zinc-Regulated Peptidase Maintains Cell Wall Integrity during Immune-Mediated Nutrient Sequestration. *Cell Rep.* 26, 2009–2018.e6 (2019)).

Instead of verifying the presence of pAB5 in recovered bacteria by PCR, as suggested by the reviewer, plasmid presence was confirmed by plating recovered bacteria on both LB-agar and LB-agar containing kanamycin and gentamycin, whose resistance is encoded on pAB5. These antibiotics effectively inhibited growth of UPAB1 without pAB5. Using this tandem plating protocol, we appreciated no pAB5 loss in our *in vivo* assays. We clarified this point in the manuscript, lines 177-179.

9. Tables S4-S5, why not provide the fold change results for each differentially expressed protein?

We thank the reviewer for this comment. We apologize for the confusion. The fold changes are provided within the supplementary tables 7, 8 and 9, in columns “N: Student's T-test Difference P- Static_P5 Static_ttest_P- Static_versus_pAB5_static” and “N: Student's T-test Difference minus_plusminus_vs_plus”. As the LFQ values are presented as log₂ values, taking the difference of these means yields the fold change. Although the confusion is understandable we would like to note this is the default output from the Perseus analysis platform (Tyanova S et al Nat Methods. 2016 Sep;13(9):731-40. doi: 10.1038/nmeth.3901) and a common way that proteomic data is presented. To avoid potential confusion we have added the following statement to the supplementary table legend “The fold change difference in the mean log₂(LFQ) value of biological conditions is provided as the difference between condition while the t-test p-value is presented as the -log₁₀ value.” Also, we added a column with the fold change difference values of the selected proteins presented in supplemental tables 2, 3 and 4.

Minor comments:

10. The Results section contains a number of sentences on the methods and can be more concise. The Discussion section is somewhat brief and superficial.

We attempted to address the reviewer's comments in the body of the text. However, due to various methods employed throughout the manuscript, we feel readability is improved by mentioning important methods points throughout the text.

Reviewer #2 (Remarks to the Author):

In this study the authors present a novel, previously not observed mechanism in *Acinetobacter baumannii* isolates, enabling or enhancing colonisation in the urinary tract.

Comments:

Uropathogens exhibit specific mechanisms, which enable them to survive and grow in urine. *A. baumannii* up to now is a rarely reported isolate in urinary tract infections. The

authors have performed a retrospective analysis of their databank revealing that approximately 20% of *A. baumannii* isolates are cultured from the urinary tract. **The statement that *Acinetobacter baumannii* is a leading source of CAUTI however needs further proof.** In most bacterial spectrum analyses of UTI or CAUTI *A. baumannii* is reported in less than 2% of the entire bacterial spectrum.

Dr. Wagenlehner is correct in asserting that global surveillance studies report that *A. baumannii* is a minor cause of UTI, so we cannot claim that *A. baumannii* is one of the principle uropathogens *overall*. However, there are multiple regional and single center reports of *A. baumannii* being the principle cause of ICU MDR CAUTI, likely from local outbreaks and/or antimicrobial practices (references in the text). This demonstrates that *A. baumannii* can have a large clinical impact as a uropathogen, despite *A. baumannii* playing a small role in UTI overall (which is dominated by Enterobacteriaceae). We modified the sentences in lines 103-105 for greater clarity.

It would therefore be interesting to see the figure of the authors epidemiological study. What was the percentage of *A. baumannii* of all urinary isolates identified in their study?

Our retrospective epidemiologic analysis and systematic literature review were designed to describe the spectrum of *A. baumannii* disease, specifically. They were not designed to ask the question of what percentage of a disease type are caused by *A. baumannii*. We want to highlight the capability of *A. baumannii* to be a uropathogen, not the epidemiological burden of *A. baumannii* urological disease. Unfortunately, none of the reports included in the literature review include the data necessary to answer the reviewer's question.

The references cited mainly describe *A. baumannii* in specific situations such as patients in ICU. **Do the authors have an explanation from their own work, what might be the driving force for this.** Is it the high antibiotic resistance, enabling *A. baumannii* to be selected during antibiotic treatment in intensive care patients, or are there additional factors.

There are very few studies directly addressing the epidemiology of *A. baumannii* urinary infections. Though *A. baumannii* has an important role in ICU-related UTIs, there is evidence that *A. baumannii* can occur in healthier patient populations. For example, the Spanish study referenced in the manuscript found that up to 30-40% of *A. baumannii* UTI occur in non-catheterized patients. In our retrospective analysis currently under review, we found that the majority of *A. baumannii* urinary isolates are isolated in the ambulatory setting or within 48 hours of hospitalization, i.e., unlikely to be hospital-acquired. Though drug resistance and critical illness/immunosuppression in the host likely play a role in ICU-related UTIs, there are likely other factors that have not been identified. Addressing these questions would require more expansive analysis of clinical cases, beyond the scope of the current study.

The isolated strain UPAB1 was grown in a patient from Argentina. **Do the authors have information about the severity of the UTI of this patient? Was it urosepsis, or asymptomatic bacteriuria or something else?**

Unfortunately, we have limited access to clinical data concerning UPAB1. We do know that UPAB1 was isolated from an ambulatory female who presented with chief complaints of dysuria and pyuria on urinalysis. We do not know the vital signs or other lab results at time of presentation, so we cannot address whether she was septic. However, given her admission course, she was not in septic shock. Thus, we graded this case as an uncomplicated UTI. We modified the sentence on lines 114-115 for clarity.

In this direction, do the authors have information about the different types of UTI found in their epidemiological study? This could answer the issue about pathogenicity of *A. baumannii* in UTI from an clinical epidemiological point of view. We agree with Dr Wagenlehner that the clinical context of urinary isolates is essential for establishing pathogenicity. Analyzing the clinical characteristics of our cohort is beyond the scope of the current study, but in-depth analysis of *A. baumannii* UTI in BJC is underway and will be published in the future. We can say that our preliminary analysis demonstrates that >50% of cases with *A. baumannii* urinary isolates are associated with symptomatic bacteruria or signs of systemic infection attributable to urinary infection.

The authors robustly describe the role of the plasmid pAB5. **Do they have information about, in how many strains of *A. baumannii* plasmid pAB5 has been identified?**

In addition is the plasmid pAB5 genetically identical or comparable to plasmid found in other typical uropathogens such as *E. coli* or other enterobacteria.

pAB5 belongs to the recently described Large Conjugative Plasmid (LCP) family (Di Venanzio, G. et al. Multidrug-resistant plasmids repress chromosomally encoded T6SS to enable their dissemination. *Proc. Natl. Acad. Sci. U. S. A.* 116, 1378–1383 (2019)).

Although highly related, none of the LCPs available in the NCBI database are identical. We identified at least 100 *A. baumannii* strains carrying LCPs in NCBI database.

However, given the large number of draft sequences obtained by Illumina-based methods (which cannot readily discern between plasmid and genomic DNA) in NCBI, and possible plasmid loss during DNA preparations, we believe several additional strains contain unidentified LCPs. Furthermore, plasmid *prevalence* cannot be accurately ascertained without a systematic molecular analysis of a defined bacterial population, which is beyond the scope of this study.

BLAST searches did not identify plasmid homologs in non-*Acinetobacter* species.

Though large plasmids carrying regulatory genes and antibiotic resistance islands do exist in other gram-negative pathogens, our extensive literature search did not identify other plasmids associated with differences in urovirulence.

F. Wagenlehner

Reviewer #3 (Remarks to the Author):

The manuscript by Venanzio et al. describes an interesting phenomenon, which is the

regulation of chromosomal genes by the presence of a plasmid in *Acinetobacter baumannii*. Additionally, this regulation by the plasmids leads to a change in virulence and niche specificity. The manuscript is well structured, clearly written and concise. There are only a few things that should be clarified or extended upon.

1. The methods are in very many places greatly abbreviated and not fully described. In these cases previous literature is referenced for details. However, a lot of this literature is pay-walled. The methods should be described in enough detail to allow replication without having to go to non-open access journals. This applies to almost every method section. For example, in line 118 you mention the construction of the strain without pAB5, which is the most central piece of the methods in the whole manuscript, as all subsequent work depends on it. However, no details on strain construction are provided and the referenced publication is not open access.

We thank Dr. Kleiner for highlighting this point and we apologize for the lack of a more thorough methods description. To address this, we have provided more details to the methods section to enable others in replicating this work without needing to access additional references.

2. Line 212 and following, the 100 ug dried protein pellets seem to come out of nowhere, there are no descriptions in the whole methods part on from what and how this protein was prepared.

We thank t Dr. Kleiner for highlighting this oversight, these details have now been provided (lines 433-440).

3. Line 237, there is no description anywhere in the manuscript on how the secretome samples were prepared.

We thank Dr. Kleiner for highlighting this oversight, these details have now been provided (lines 415-422).

4. Line 253, more detail on the MaxQuant search parameters would be useful. At least FDRs for protein and peptide identification should be reported.

We thank the reviewer for highlighting this oversight. A 1% protein and peptide level FDR has been implemented which is the default setting for Maxquant. We have added the sentence "The default Maxquant FDR setting of 1% FDR at the protein and peptide levels were used." To ensure users are aware that the full Maxquant search parameters are provided within the PRIDE submission, we have also modified the methods to state "The mass spectrometry proteomics data have been deposited to the ProteomeXchange Consortium via the PRIDE partner repository with the dataset identifier PXD011302 and PXD011341 which contains all the raw MS data files and Maxquant outputs including list of all search parameters."

5. Both in the methods and the results there is no mention of the number of replicates used for proteomics and transcriptomics. Based on the proteomic raw data I am guessing that the replicate numbers were appropriate, but this needs to be clearly documented in the text.

We apologize for not being clear on these points. Four biological replicates of each conditions were used for the proteomic analysis and three biological replicates of each conditions were used for the transcriptomic analysis (line 497). To ensure this point is clear to readers we have expanded on the methods sections and added the following: Line 455 “Four biological replicates of each whole proteome condition were analyzed in a randomized manner to minimize batch effects. Wash runs were performed between replicates to prevent samples carry over.”

Line 465 “Four biological replicates of each whole proteome condition were analyzed in a randomized manner to minimize batch effects. Wash runs were performed between replicates to prevent samples carry over.”

To further clarify this point, the supplementary table legends now include statements which highlight that these comparative proteomics were undertaken with four biological replicates and also highlight which LFQ experiment corresponds to which biological replicate.

6. FDRs and replicate numbers should also be reported in figure captions for proteomic and transcriptomic data.

We thank Dr. Kleiner for their concern on the statistical conclusions of our proteomic and transcriptomic analysis. For the transcriptomic analysis, DESeq2 uses adjusted p-value instead of FDR for statistical significance. To address their concern, we have added a sentence in figure 7 legend stating that all genes in the transcriptomic volcano plot with adjusted p-values above the statistical cut-off of .1 (ie not significantly different between conditions) per the DESeq2 vignette are in grey, while the genes with adjusted p-values below .1 are blue or red.

For the proteomic analysis we have added these details on line 472 and in the Figure 6 legend.

7. The Type 6 secretion system should be mentioned in the abstract or in the keywords to increase visibility of this very important topic in search results.

We have modified the abstract as suggested.

8. The abstract is very well written. One point I think could be made more clear is the fact that removal of the plasmid leads to increased infection in the respiratory tract as this is a concrete and critical result.

We thank Dr. Kleiner for highlighting this point. We have modified the abstract as suggested

Signed: Manuel Kleiner